# Research on a Regional Landslide Early-Warning Model Based on Machine Learning—A Case Study of Fujian Province, China

Yanhui Liu [1,*], Junbao Huang [2], Ruihua Xiao [1], Shiwei Ma [3] and Pinggen Zhou [1]

1 China Institute of Geo-Environment Monitoring (Technical Guidance Center for Geo-Hazards Prevention of MNR), Beijing 100081, China
2 Fujian Monitoring Center of Geological Environment, Fuzhou 350002, China
3 Institute of Geology and Geophysics, Chinese Academy of Sciences, Beijing 100029, China
* Correspondence: lyanhui@mail.cgs.gov.cn

**Abstract:** China's landslide disasters are serious, and regional landslide disaster early-warning is one of the important means of disaster prevention and mitigation. The traditional regional landslide disaster early-warning model, however, is limited by the complex landslide induction mechanism, limited data accumulation, and insufficient big data analysis methods, and has problems such as limited early-warning accuracy and insufficient refinement. In this paper, a machine learning method was introduced into the field of regional landslide disaster warning. From the model construction process of training sample-set construction, sample learning and training, model parameter optimization, model preservation, warning output, and so on, a method for constructing a regional landslide early-warning model based on machine learning was systematically proposed. In the sample learning and training, 80% of the training sample-set was used as the training set, and 20% was used as the test set for five-fold cross validation. The Bayesian Optimization algorithm was used to optimize the model parameters, and the accuracy, ROC curve, and AUC value were used to verify the model accuracy and model generalization ability. With China's Fujian province as an example, based on nine years of geological and meteorological data (2010–2018), geological environment factors, factors of hazard-affected bodies and historical disaster situations, and rainfall-induced factors in four categories, a total of 26 indicators were used as input characteristic parameters. Six machine learning algorithms were adopted to improve model training; the results showed that the Random Forest algorithm performed the best, giving an accuracy of 92.3%, and was the model with the best generalization ability (AUC was 0.955). The second best was the Artificial Neural Network model, with an accuracy of 0.937 and an AUC of 0.935. Next were the Nearest Neighbor model, the Logistic Regression model, and the Support Vector Machine; the poorest results were from the Decision Tree model. Finally, the typical rainfall-type landslide disaster process in Fujian Province was selected as an example to verify the Random Forest algorithm model. The results showed that compared with the early-warning results of the original explicit statistical model, the hit rate of the new model was 6 times, or equal to that of the original model, and the landslide density in the early-warning area of the new model was 1.6–1.7 times that of the original model. Preliminary verification showed that the new model based on the Random Forest method has obvious advantages, a higher hit rate and a smaller warning area, and can achieve more accurate warnings. The follow-up will continue to track the new landslide disaster situation in the study area and carry out model verification and correction.

**Keywords:** landslide; early-warning model; machine learning; Random Forest; model study

## 1. Introduction

China is one of the countries with the most widespread and serious geological disasters in the world. Geological disasters spread throughout the country's mountainous and highland areas, with nearly one million known locations, causing hundreds of deaths and billions of yuan of direct economic losses every year (according to the National Geological

Disaster Bulletin issued by the Ministry of Natural Resources). More than 20 countries or regions in the world, including Hong Kong, China, the United States, Italy, Brazil, Japan, etc., have also performed or are carrying out regional geological disaster early-warning and mitigation services to varying degrees [1]. Since 2003, the Chinese mainland has carried out a meteorological early-warning for regional geological disasters and achieved good results in disaster prevention and mitigation [2–10]. Additionally, in the monitoring and early-warning demonstration area [11,12], the Three Gorges Reservoir Area [2], Wenchuan earthquake disaster area [13], and other key regions, extensive research and practice have also been carried out. According to statistics, since 2003, owing to the efforts of various parties, the number of deaths and missing caused by geological disasters has decreased from about 1000 per year during the Tenth Five-Year Plan period to about 500 since the Twelfth Five-Year Plan period, which indicates that important contributions have been made to meteorological early-warning and prediction of geological disasters [3,4].

The model study is the basic scientific problem of regional landslide early-warning. A large number of scholars have carried out long-term and unremitting research on it. The first and most widely used model is the statistics-based critical rainfall threshold model, which was systematically studied in Hong Kong, China, and the United States [14,15], and has been widely used as a reference in other countries or regions due to its advantages of simplicity and ease of generalization [16–19]. Based on the statistical principle, the explicit statistical early-warning model proposed by Liu Chuanzheng et al. [2] has been deeply explored and applied at all levels of early-warning, key research areas, and monitoring and early-warning areas in mainland China [4–7,9,11,12]. A regional dynamic early-warning model based on the mechanism process analysis of rainfall–seepage and disaster occurrence has been continuously studied. By coupling slope stability analysis with a hydrogeological model, the critical rainfall index for landslide initiation has been determined [20–23], and the physical significance of the dynamic early-warning model is clear, but due to the complex parameter input and uncertainty in the model, this method is mostly used in the small-scale research process, and its practical operation is also limited.

In recent years, with the rapid development of artificial intelligence technology, machine learning and deep learning algorithms based on artificial intelligence have become increasingly mature and widely used in various industries. In the field of geological disaster prevention and control, a variety of machine learning algorithms have been widely used in landslide spatial evaluation and prediction in recent years, such as Artificial Neural Networks, Decision Trees, Support Vector Machines, Random Forests, etc. [24–29]. Most of the above-mentioned studies introduce machine learning algorithms into landslide spatial evaluation and prediction to evaluate regional landslide sensitivity or susceptibility. After the spatial evaluation is completed, the critical precipitation threshold is determined by traditional statistical methods [30]. However, there are few related achievements of realizing spatial and temporal warning for regional landslide disasters based directly on a machine learning algorithm.

Aiming at the problems that exist in the traditional regional landslide early-warning models, such as complex landslide-inducing mechanisms, limited data accumulation, and insufficient data analysis methods, past results lead to limited warning accuracy and a lack of indicator precision. Through the training sample-set construction, comparative analysis of various machine learning methods could improve the precision of early-warning models and other aspects of the research to solve these problems. This paper systematically expounded on the construction method for a regional landslide disaster early-warning model based on crucial components of machine learning algorithms: training sample-set construction, model training, optimization evaluation, and early-warning modeling. Taking Fujian Province of China as an example, the Random Forest algorithm and Nearest Neighbor algorithm were applied, and Support Vector Machine, Logistic Regression, Decision Tree, Artificial Neural Network, and other algorithms, based on the geological and meteorological data from 2010 to 2018, were used to construct a regional landslide disaster early-warning model for Fujian Province. We selected two typical rainfall-induced

landslide disasters in Fujian Province in 2019. Taking the process as an example, we carried out the live operation and application verification of the early-warning model.

## 2. Geological Background of the Study Area

Fujian Province is located in a mountainous highland area along the southeast coast of mainland China. It is one of the provinces with complex terrain, abundant rainfall, and frequent landslide disasters. At the present, the forest coverage rate of Fujian Province rate has reached 66.8 percent, ranking first in China for 43 consecutive years. The highlands and mountainous areas of the study area account for about 80% of the total area, mainly in the central and western regions of Fujian Province. The terrain of Fujian Province is generally high in the northwest and low in the southeast. Under the control of the New Cathaysia tectonic system, the western Min-Dashan belt and the middle Min-Dashan belt are formed in the west and central part of Fujian Province, and there are disconnecting valleys and basins between the two belts. There are many river systems in Fujian Province, and their flow direction is mostly from west to east. The rivers are mostly mountainous rivers with abundant water volume, great seasonal changes, and rapid flow. The province has a subtropical humid monsoon climate, with an average annual rainfall of 1000~1900 mm, abundant rainfall, strong monsoon circulation, and a remarkable monsoon climate. Therefore, Fujian is an area prone to natural disasters, frequently-occurring areas, and worst-hit areas, especially landslide disasters are most widely distributed (Figure 1). The residual slope soil layer in the study area is widely developed. The slope gradient is generally 0~30°, locally greater than 30°. Due to the poor geological environment conditions, most of the villages in the mountainous area were built on the slopes, forming a large number of high and steep slopes in front of or behind the houses. During the flood season, sudden geological disasters such as collapses and landslides occur frequently. Although the scale of landslides is small, most of them occur in front of or behind residents' houses, which can easily cause casualties and property losses. The occurrence of landslide disasters in Fujian Province is closely related to the terrain and induced by heavy rainfall and human engineering activities. The disaster-vulnerable terrain is prone to occur on gradients with a slope of more than 20°; generally, the hazards are liable to happen during the heavy rainfall period from May to June and the typhoon and rainy period from July to September, and they usually take place in front of and behind houses sections such as the cutting slopes, planting economic forests, and spoiling soil along the slopes [31].

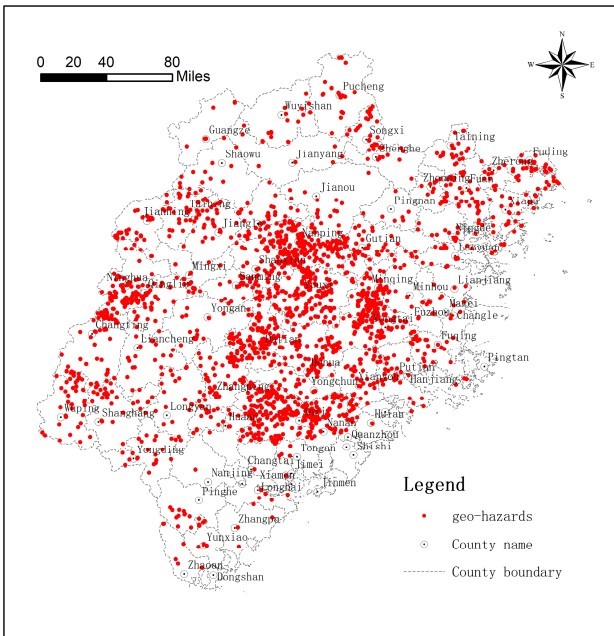

**Figure 1.** Distribution map of geological disaster points in Fujian Province.

## 3. Determination of Evaluation Index and Selection of Algorithm

### 3.1. Extraction of Evaluation Indexes

Landslide disasters are influenced and affected by the combination of topography, geological conditions, environmental conditions, and human engineering. Analyzing influence factors of landslide disasters is the basis for carrying out mechanism research, early warning and forecasting, and disaster prevention and mitigation. According to the existing survey and monitoring data in the study area, as well as the development and distribution of regional landslide disasters and the analysis of influencing factors [31–33], the input features of influencing factors are extracted from four aspects: geological environment factors, hazard-affected body factors, historical disasters, and rainfall-induced factors. Geological environment factors are the geological environment background factors of landslide disasters in the study area, which determine the potential susceptibility degree of landslide disaster in this area. Hazard-affected body factors are the evaluation indexes of the hazard-affected body that may be caused when a landslide occurs. Historical disasters refer to the number of historical landslide disasters in the study area. Rainfall-induced factors are the direct rainfall-inducing factors of landslides in the forecast period. These four categories cover all the geological environment aspects that influence the prediction of the possibility and risk of landslides disaster in this study area.

The extracted geological environment factors mainly include grade, geomorphic type, stratigraphic lithology, annual rainfall, vegetation type, water system influence, etc. Based on the 1:200,000 and 1:500,000 geological environment and geological disaster survey database of Fujian Province, the hourly precipitation data of Fujian Province from 2010 to 2018 (nearly 2000 stations), and the grid precipitation Real-time (QPE) data of Fujian Province in 2021 (grid scale: 5 km × 5 km), the correlation analysis between these six geo-environmental factors evaluation indexes and the spatial distribution of landslide hazards in the region was carried out. According to the results of correlation analysis, the grade factors were classified into five categories: 0~15°, 15~25°, 25~35°, 35~50°, ≥50° (Figure 2a); the geomorphic types were classified into five categories: plains, hills, low mountains, medium mountains, and high mountains (Figure 2b). The lithological factors of the formation were classified into a massive hard granite rock group, hard-harder diorite rock group, massive hard-harder tuff lava rock group, medium-thick layer, and relatively hard sandstone rock group, thin layer soft mudstone, shale rock group, medium-thick hard quartz gneiss rock group, medium-thick hard carbonate rock group, and loose sandy clay soil layer (Figure 2c). The annual average rainfall types were classified into 13 categories such as 1400–1450 mm, 1450–1500 mm, and so on, up to >2000 mm (Figure 2d). Vegetation affects landslides through coverage, density, abundance, height, underground biomass, leaf area index, and aboveground biomass. According to the correlation between the spatial distribution of landslides and the distribution of vegetation types, the vegetation types were classified into six types (Figure 2e); the water system was classified into two categories based on whether the impact distance is greater than 500 meters (Figure 2f). Detailed information is shown in Table 1.

The main indicators of hazard-affected body factors extracted are roads, houses, and population density. The road distribution layer was extracted from 1:250,000 DLG data, and the distance from the road 0~500 m and ≥500 m were classified into two categories and quantified separately (Figure 3a). The house distribution layer was extracted from 1:250,000 DLG data, and the distances from the house 0~500 m and ≥500 m were classified into two categories and quantified separately (Figure 3b). According to the data of the sixth National Population Census, the population density (unit: pieces /km$^2$) types were classified into seven categories, 50–100, 100–150, 150–300, 300–450, 450–600, 600–750, >750, and quantified, respectively (Figure 3c). The specific content is shown in Table 1.

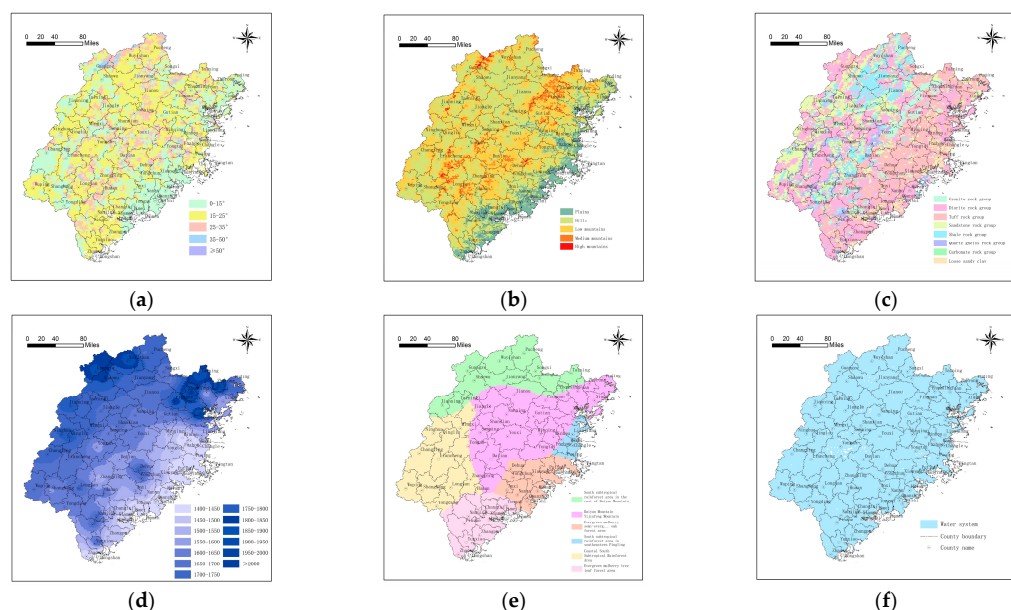

**Figure 2.** Geological environment factor index of the study area: (**a**) grade, (**b**) geomorphic type, (**c**) formation lithology, (**d**) annual rainfall, (**e**) vegetation type, (**f**) water system.

**Table 1.** Input features and parameters of the training sample set.

| Category | Serial Number | The Input Features | Characteristic Parameters | Data Sources and Processing Methods |
|---|---|---|---|---|
| Geological environment factors | 1 | Grade/(°) | ① 0~15; ② 15~25; ③ 25~35; ④ 35~50; ⑤ ≥50 | 1:100,000 grade map of Fujian Province, classification and quantification |
| | 2 | Geomorphic type | ① plains; ② hills; ③ low mountains; ④ medium mountains; ⑤ high mountains | 1:200,000 geomorphic type map of Fujian Province, classification and quantification |
| | 3 | Stratigraphic lithology | ① massive hard granite rock group; ② hard-harder diorite rock group; ③ massive hard-harder tuff, tuff lava rock group; ④ medium-thick layer and relatively hard sandstone rock group; ⑤ Thin layer soft mudstone, shale rock group; ⑥ medium-thick hard quartz gneiss rock group; ⑦ medium-thick hard carbonate rock group; ⑧ loose sandy clay soil layer | 1:200,000 stratigraphic lithology map of Fujian Province, classified and quantified |
| | 4 | Annual rainfall/(mm) | ① 1400–1450; ② 1450–1500; ③ 1500–1550; ④ 1550–1600; ⑤ 1600–1650; ⑥ 1650–1700; ⑦ 1700–1750; ⑧ 1750–1800; ⑨ 1800–1850; ⑩ 1850–1900; ⑪ 1900–1950; ⑫ 1950–2000; ⑬ >2000 | 1:500,000 geological disaster survey and zoning reports in Fujian Province, vectorization acquisition, classification, and quantification |
| | 5 | Vegetation type | ① South subtropical rainforest area in the east of Daiyun Mountain; ② Daiyun Mountain Yijiufeng Mountain Range; ③ Evergreen mulberry-semi-evergreen oak forest area; ④ South subtropical rainforest area in southeastern Pingling; ⑤ Coastal South Subtropical Rainforest Area; ⑥ Evergreen mulberry tree leaf forest area | 1:500,000 vegetation type map of Fujian Province, classification and quantification |
| | 6 | Water system influence/(m) | ① 0~500; ② ≥500 | 1:500,000 water system distribution map of Fujian Province, calculation buffer classification quantification |

| Category | Serial Number | The Input Features | Characteristic Parameters | Data Sources and Processing Methods |
|---|---|---|---|---|
| Hazard-affected body factors | 7 | The distance from road/(m) | ① 0~500; ② ≥500 | The road distribution layer was extracted from 1:250,000 DLG data, and the buffer classification quantization was calculated |
|  | 8 | The distance from house/(m) | ① 0~500; ② ≥500 | The house distribution layer was extracted from 1:250,000 DLG data, and the buffer classification and quantification were calculated |
|  | 9 | Density of population/ (pieces/km²) | ① 50–100; ② 100–150; ③ 150–300; ④ 300–450; ⑤ 450–600; ⑥ 600–750; ⑦ >750 | The sixth national population census data, classification and quantification |
| Historical disaster factors | 10 | Historical disaster/(pieces) | the number of historical damage points of each grid cell/10 | Potential points of landslides in 1:500,000 geological disaster survey data in the study area, National Geological Disaster Database (2010–2018), with a scaled range |
| Rainfall-inducing factors | 11 | Rainfall of that day/(mm) | actual rainfall value/10 | The data of meteorological and water conservancy hourly precipitation stations from 2010 to 2018 were interpolated and the range was scaled |
|  | 12 | Rainfall in the previous day/(mm) | actual rainfall value/10 |  |
|  | 13 | Rainfall in the last two days/(mm) | actual rainfall value/10 |  |
|  | ... | ... | actual rainfall value/10 |  |
|  | 26 | Rainfall in the last 15 days/(mm) | actual rainfall value/10 |  |

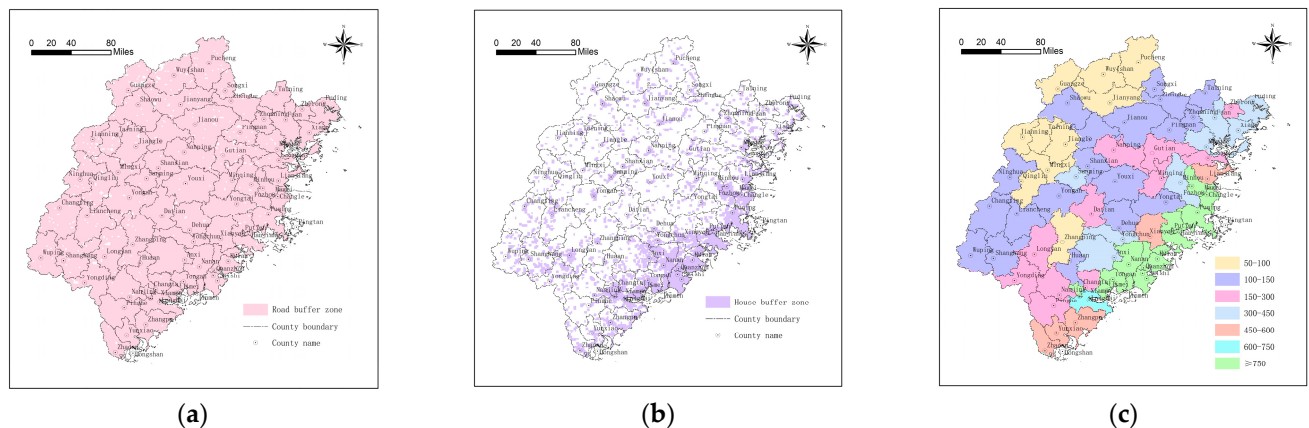

|  |  |  |
|:--:|:--:|:--:|
| (**a**) | (**b**) | (**c**) |

**Figure 3.** Hazard-affected body factors: (**a**) distance from the road, (**b**) distance from the house, (**c**) population density.

The historical landslides samples were extracted from the 1:500,000 geological disaster survey data of the study area and the national geological disaster database (2010–2008), and the scope was scaled by ten times of reduction (Figure 1).

The extracted index of rainfall-inducing factors mainly considers the rainfall of the current day and the daily rainfall of the previous 15 days. According to the data of

meteorological and water conservancy hourly precipitation stations from 2010 to 2018, the index of rainfall-inducing factors of each grid cell were extracted, and the range was realized by reducing it ten times.

Through the above analysis, the twenty-six input characteristics of four categories of input characteristic factor indexes (geological environment factor, hazard-affected body factor, historical disaster situation, and rainfall-inducing factor) in Fujian Province were obtained. Among them, nine input characteristics of two categories of geological environment factors and hazard-affected body factors were classified and quantified. Data range scaling was carried out for seventeen input features of two categories: historical disaster conditions and precipitation-inducing factors, as shown in Table 1.

### 3.2. Selection of Machine Learning Algorithms

Commonly used machine learning algorithms mainly solve three major problems of regression, clustering, and classification. The prediction data of the regression algorithm is continuous numerical data, mainly including linear regression, logistic regression, etc., in which logistic regression can also solve the classification problem. The prediction data of the clustering algorithm is categorical, and the category is unknown, which is mainly a K-means clustering algorithm. The prediction data of the classification algorithm is classified data, and the category is known. The main algorithms include the Nearest Neighbor algorithm, Support Vector Machine, Artificial Neural Network, Logistic Regression, Decision Tree, and Random Forest. For regional landslide disaster warning problems, the machine learning classification algorithm is mainly used. In this paper, six algorithms, including the Random Forest algorithm, Nearest Neighbor algorithm, Support Vector Machine, Logistic Regression, Decision Tree, and Artificial Neural Network were selected to establish a landslide early-warning model. The comparative analysis of algorithms is shown in Table 2.

**Table 2.** Comparison and analysis of commonly used machine learning classification algorithms.

| Commonly Algorithm | Principle | Advantages | Disadvantages |
|---|---|---|---|
| Logistic Regression | Based on the existing data, a regression formula (the best-fitting parameter set) was established for the classification boundary. | ① The calculation is small and the speed is fast; ② It has a good probability explanation; ③ Can easily update the model. | ① It is easy to under-fit and the accuracy is limited; ② It can only deal with two classification problems, and it must be linearly separable. |
| Nearest Neighbor algorithm | Classification is carried out by measuring the distance between different eigenvalues. | ① Suitable for multiple classification problems; ② High accuracy; ③ It is not sensitive to abnormal points. | ① Large amount of calculation, poor comprehension; ② When the training data is highly dependent and the samples are unbalanced, the prediction accuracy of rare categories is low. |
| Decision Tree | The tree structure is used to establish the decision model according to the data attribute. | ① Easy to explain and explain, good at dealing with the interaction between features; ② Suitable for analyzing discrete data; ③ Small-scale data sets are effective. | ① Poor treatment of continuous variables; ② Online learning is not supported, and the Decision Tree needs to be reconstructed when there are new samples. ③ Easy to overfit. |

**Table 2.** *Cont.*

| Commonly Algorithm | Principle | Advantages | Disadvantages |
| --- | --- | --- | --- |
| Artificial Neural Network | Simulating biological neural networks, a class of pattern-matching algorithms is a huge branch of machine learning with hundreds of different algorithms. | ① High classification accuracy; ② Strong learning ability. | ① A large number of parameters are required; ② Unable to observe the learning process, the results are difficult to interpret; ③ The study time is long. |
| Support Vector Machine | To find the optimal hyperplane, the data can be divided into two parts, each part of the data belongs to the same class. | ① It can solve nonlinear classification; ② The idea of classification is simple. | ① Large memory consumption, difficult to implement for large-scale training samples; ② It is difficult to solve the multi-classification problem; ③ Difficult to explain, complex to run and optimize. |
| Random Forest algorithm | A forest is built randomly. The forest is composed of many independent Decision Trees, and finally, the optimal classification result is obtained comprehensively. | ① The limited sample can be fully applied; ② It has the advantages of diversity and accuracy. | It will overfit on some noisy problems. |

The quality of a machine learning model depends on its evaluation accuracy and model generalization ability. Several common parameters for model evaluation include the following:

(1) Accuracy (ACC), which expresses the model evaluation accuracy. The accuracy of the model is the ratio of the number of samples correctly predicted by the model to the total number of samples. In addition, there are metrics such as precision, recall, and F1 value.

(2) The ROC curve and AUC value express the generalization ability of the model. ROC (Receiver Operating Characteristic) curve refers to the receiver operating characteristic curve, which is a comprehensive index reflecting the continuous variables of sensitivity and specificity. Its main analysis tool is a curve drawn on a two-dimensional plane; AUC (Area Under roc Cure) value is the area under the ROC curve. Usually, the value of AUC is between 0.5 and 1.0, and the larger the AUC value, the better the performance of the model.

(3) Learning curve, which describes the model fitting problem and judges whether the model is over-fitting or under-fitting.

## 4. Regional Landslide Early-Warning Model Method Based on Machine Learning

### 4.1. Construction of Training Sample-Set

The accurate construction of the training sample-set is the foundation of the machine-learning regional landslide disaster warning model, and it directly determines the accuracy and generalization ability of the warning model to a certain extent. The occurrence of regional landslide hazards is controlled by the coupling effect of geological environmental conditions and rainfall conditions. From the perspective of input features, machine learning samples include attributes of three aspects: geographic location, geological environmental conditions, and precipitation conditions. Geographical location refers to the spatial geographic coordinates of the point where the sample is located; geological environmental conditions refer to the geological environmental background conditions of the sample; precipitation conditions refer to the induced rainfall factors of the sample. From the perspective of output features, the training sample-set includes positive samples (landslide points, generally denoted as one) and negative samples (non-landslide points, generally

denoted as zero). The balance of the number of positive and negative samples should be considered when sampling positive and negative samples.

The construction process of the training sample-set is shown in Figure 4, which mainly includes three steps: geological environment and rainfall factor feature library construction, positive and negative sample sampling, and sample feature attribute extraction.

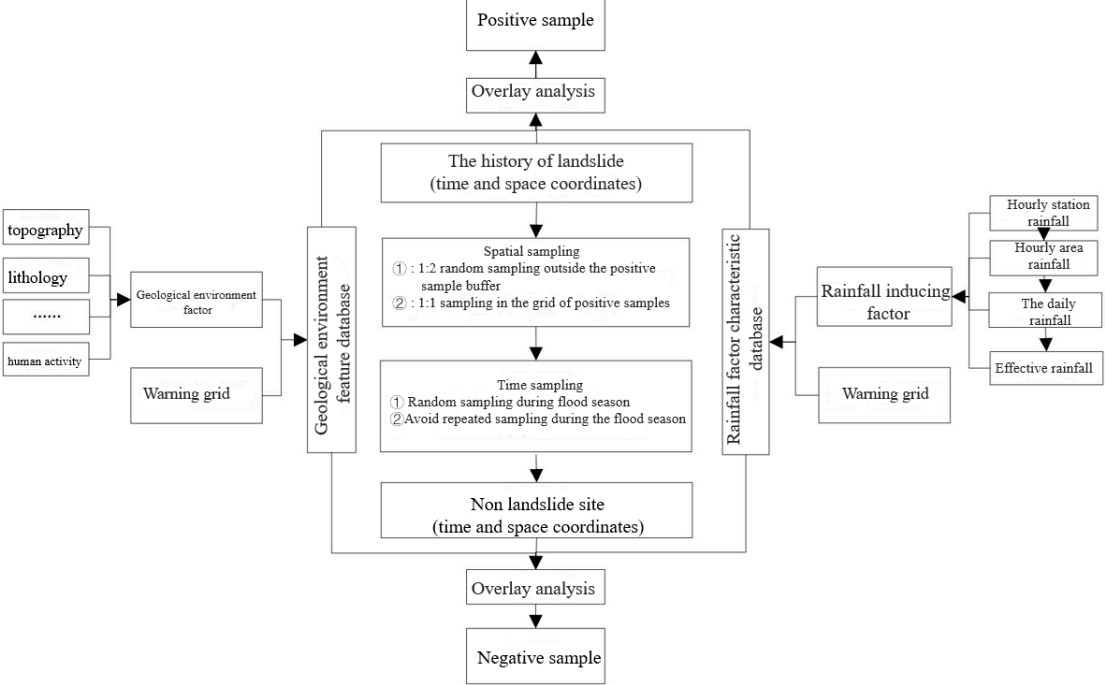

**Figure 4.** Training sample-set construction flowchart.

### 4.1.1. Construction of Characteristic Database of Geological Environment and Rainfall Factors

The construction of the geological environment characteristic database and rainfall factor characteristic database is completed based on the analysis of regional landslide disaster distribution patterns and influencing factors in the study area. The geological environmental factors affecting the occurrence of regional landslides generally include topography, strata lithology, human activities, etc.; the rainfall-inducing factors affecting the occurrence of regional landslides generally include daily rainfall, previous rainfall, previous effective rainfall, etc.

The geological environment factors and rainfall-inducing factors are overlaid with the warning grid profiling unit respectively (Figure 5), and the geological environment characteristic library and rainfall factor characteristic library of the early-warning grid unit are obtained. The geological environment characteristic library contains the characteristics attributes of each geological environment influencing factor of each early-warning grid unit; the rainfall factor characteristic library contains the daily rainfall characteristic attribute or effective rainfall characteristic attribute of each early-warning grid unit.

The early-warning grid unit is determined according to the size of the study area and the actual need for early-warning. It is generally a uniform grid unit, which can be determined by referring to the early-warning space accuracy in the relevant early-warning standards (Table 3).

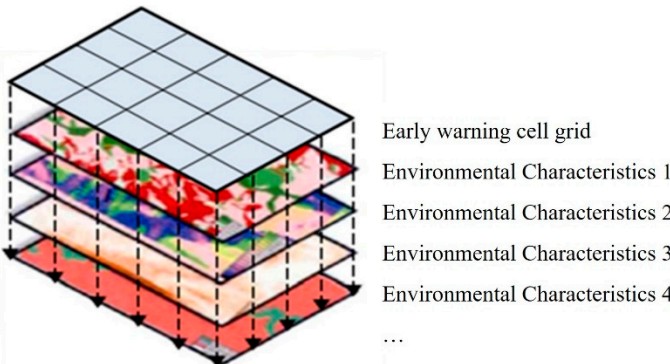

**Figure 5.** Schematic diagram of spatial superposition analysis.

**Table 3.** Precision division of early-warning space (reproduced from [34]).

| Warning Grading | Recommended Early-Warning Accuracy |
| --- | --- |
| At the national level | 1:500,000–1:1,000,000 |
| At the provincial level | 1:100,000–1:500,000 |
| The municipal | 1:50,000–1:100,000 |
| At the county level | 1:10,000–1:50,000 |
| Thematic early-warning area | 1:2000–1:10,000 |

### 4.1.2. Sampling Method of Positive and Negative Samples

Positive samples refer to the points where landslides occur, and the sampling of positive samples is generally based on historical landslide data. The screening requires that landslides should have both definite spatial geographic coordinates (the specific accuracy is determined by the specific conditions of the study area) and time coordinates (generally accurate to the day in the 24-h early-warning). Generally speaking, the sampling accuracy of positive samples is higher.

Negative samples refer to points where landslides do not occur, which cannot be obtained directly. In this paper, the negative samples were sampled in two aspects based on the method of "random sampling under space-time constraints" [35,36]: first, the size of the buffer radius was corrected; secondly, in addition to random sampling outside the positive sample buffer, sampling was supplemented in the grid where the positive sample is located, that is, the negative samples were completed in two parts. The schematic diagram is shown in Figure 6.

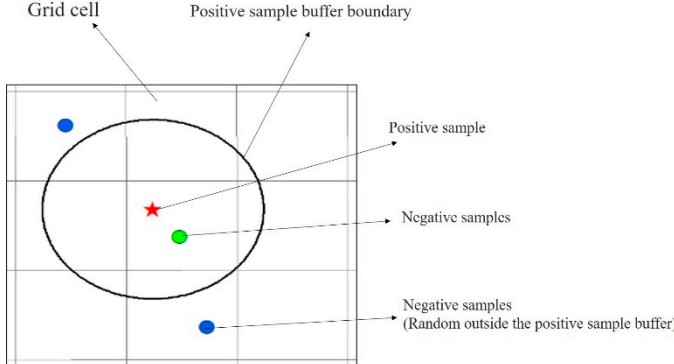

**Figure 6.** Schematic diagram of negative samples space sampling based on positive samples.

(1)　Collect negative samples outside the positive samples buffer

Determination of the spatial location of the negative samples, random sampling in the space outside a certain buffer zone of positive samples, and the determination of the radius of the buffer zone should take into account the minimum early-warning grid cell size in the study area and the distribution of historical landslide points. In this paper, the buffer radius was used as the warning grid unit size to ensure the balance of positive and negative samples; the number of negative samples collected was two times that of the positive samples.

Time attribute assignment of negative samples: under the constraints of a certain period (according to the completeness of rainfall data in the study area, it is generally the whole period of the multi-year flood season), the random function is adopted for sampling. The random function is as follows:

$$T = \text{RAND}(T_1, T_2) \tag{1}$$

Description:

T, time acquired randomly;

$T_1$, the lower limit of the time range of random acquisition time;

$T_2$, the upper limit of the time range of random acquisition time.

(2)　Negative samples are collected in the grid where positive samples are located

For the determination of the spatial location of negative samples, random sampling was performed in the grid where the positive samples are located. The number of negative samples collected was about equal to the number of positive samples.

Time attribute assignment of the negative sample also uses the random function shown in Equation (1), with the additional constraint that the negative sample time property sampled should be different from the positive sample.

4.1.3. Feature Attribute Extraction and Data Screening

The positive and negative samples are spatially overlaid with the geological environment feature database to extract the geological environment feature attributes of the positive and negative samples. Based on the temporal attributes of the positive and negative samples, and the rainfall factor feature attributes of the positive and negative samples were extracted by the query. At this point, the construction of the whole training sample-set was completed. In addition, data cleaning is particularly important throughout the training sample-set construction process, and the model evaluation accuracy is higher using the cleaned data set. Data cleaning generally consists of two categories:

(1)　Handling data errors: for example, manual errors, data transmission errors, equipment failures, and ambiguity of geological information can affect the original data set, these errors data must be processed and cleaned in advance. In general, this type of data cleaning refers to the imputation or elimination of missing values and the identification of outliers in the data.

(2)　Feature attribute preprocessing: considering the dimension difference of the input features of the training samples, it is necessary to perform uniform normalization or feature scaling on the input features of the samples. Different machine learning algorithms differ in their sensitivity to the difference of input feature scales, and the requirements for input feature attribute preprocessing are also different. It is recommended that the input features of the training samples be uniformly normalized or scaled before model training. Generally, the range of input features of the samples should be at least not much different; otherwise, it will directly affect the accuracy of the model.

In summary, the above-mentioned training sample-set construction method was used to complete the positive and negative samples in Fujian Province, and the sample-set covers 15,589 samples in the past nine years (2010–2018). Among them, there are 3562 positive samples and 12,027 negative samples, and the ratio of positive and negative samples is about 1:3.4. The spatial distribution of positive and negative samples is shown in Figure 7.

The positive and negative sample attributes determine the output features of the final training sample-set. The output features of positive samples were taken as one, and the output features of negative samples were taken as zero.

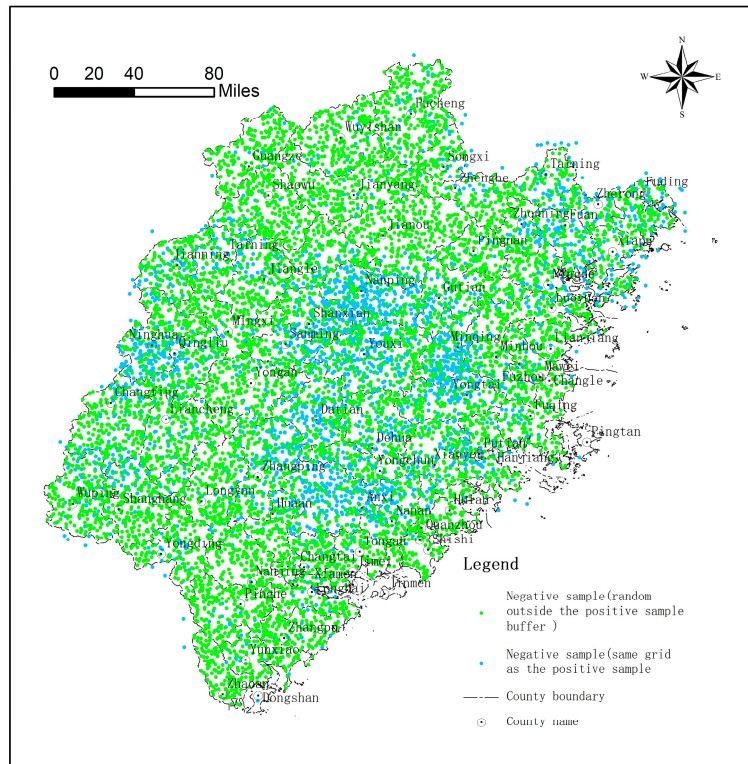

**Figure 7.** Spatial distribution map of negative samples.

### 4.2. Optimization of Model Parameters

The selection of model parameters has a significant impact on model accuracy, and model optimization is crucial in model construction. The most commonly used model parameter optimization methods include traditional methods and hyperparametric Optimization algorithms. The traditional parameter optimization method is the grid search method, whose optimization accuracy is inversely proportional to its speed. In order to optimize parameters more efficiently, the Bayesian Optimization algorithm has gradually emerged [37,38]. The Bayesian Optimization algorithm adopts the Gaussian process to fit the distribution of the objective function by increasing the number of samples, and the objective function is optimized by cross-verifying the accuracy. Each iteration outputs a hyperparameter, and the hyperparameter is optimized in the process of finding the optimal value.

### 4.3. Model Saving and Early-Warning Output

The model trained by machine learning can be saved by calling the DUMP function in the python environment. The model is generally saved as a .pkl format file.

When the actual early-warning is running, the trained early-warning model is directly called by the LOAD function to output the probability of possible landslide disasters. According to the probability, the warning level is determined by classification. The setting of graded breakpoints can refer to the classification table of geological disaster meteorological risk early-warning grades (according to the geological disaster regional meteorological risk early-warning standard (trial) (T/CAGHP 039-2018)), and can also be fine-tuned according to the specific conditions of the study area. Considering that the output threshold was set to 0.5 in the machine learning algorithm, the geohazard meteorological warning probability class classification was adjusted in conjunction with the specific conditions of the study area. That is, when the output probability P $\geq$ 50% and P < 60%, the yellow warning of

landslide disaster is issued; when the output probability P ≥ 60% and P < 80%, the orange warning of landslide disaster is issued; when the output probability P ≥ 80%, a red warning of landslide disasters is issued, as shown in Table 4.

**Table 4.** Division of early-warning levels.

| The Warning Level | The Risk of Landslides | Output Probability/P |
|---|---|---|
| Red alert | highest risk | P ≥ 80% |
| Orange alert | higher risk | 60% ≤ P < 80% |
| Yellow warning | high risk | 50% ≤ P < 60% |

## 5. Result and Verification

### 5.1. Model Parameter Optimization Training and Effect Evaluation

The training sample-set of regional landslide early-warning in Fujian Province was divided into the training test sets in the ratio of 4:1. The parameters were optimized by a Bayesian Optimization algorithm and five-fold cross-validation. The six commonly used machine learning classification models were compared and evaluated to compare the accuracy and model generalization ability indexes of each model. The optimization parameters and effect evaluation comparison of the six models is shown in Table 5 and Figure 8.

**Table 5.** Comparison of partial hyperparameter optimization and model evaluation of six machine learning algorithms.

| Machine Learning Model | Accuracy | Model Generalization Ability | Hyperparameter | Hyperparameter Value |
|---|---|---|---|---|
| Random Forest algorithm | 0.923 | 0.955 | n_estimators | 118 |
| | | | max_depths | 10 |
| | | | min_samples_split | 3 |
| Nearest Neighbor algorithm | 0.932 | 0.924 | n_neighbors | 10 |
| Decision Tree | 0.937 | 0.904 | max_depths | 4 |
| Support Vector Machine | 0.932 | 0.920 | C | 3 |
| | | | gamma | 0.003 |
| Logistic Regression | 0.940 | 0.922 | C | 5 |
| Artificial Neural Network | 0.937 | 0.935 | hidden_layer_sizes | (6,7) |
| | | | max_iter | 1680 |

According to the calculation results of the six machine learning algorithms (Table 5), it can be seen that the Random Forest model had the best performance; its accuracy rate was 0.923, the model generalization ability was the best (AUC = 0.955), and the model had no overfitting phenomenon; the learning and ROC curves are shown in Figure 9. The second is the Artificial Neural Network model, with an accuracy rate of 0.937 and an AUC of 0.935, followed by the Nearest Neighbor model, Logistic Regression model, and Support Vector Machine model, with AUCs of 0.924, 0.922, and 0.920, respectively; the worst was decision-making tree, its AUC value being 0.904 and its accuracy 0.937.

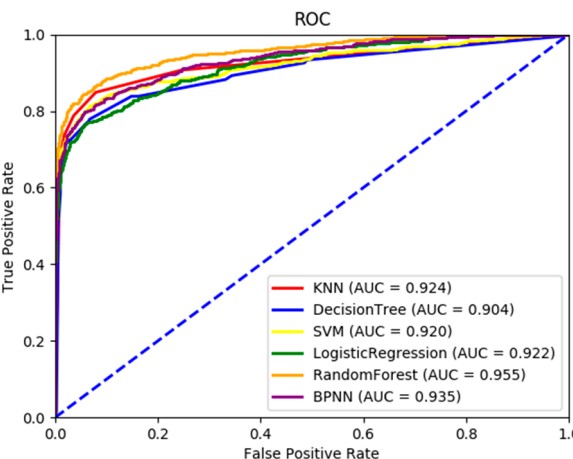

**Figure 8.** Comparison of ROC and AUC of six machine learning early-warning models.

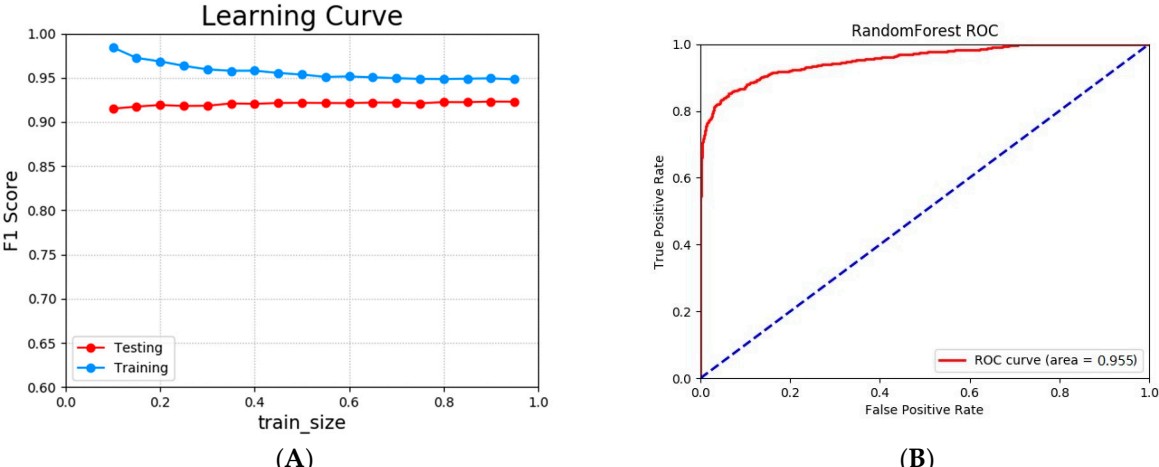

**Figure 9.** Learning curve (**A**) and ROC curve (**B**) of the Random Forest early-warning model.

*5.2. Model Early-Warning Verification—Taking the Random Forest Algorithm as an Example*

On two days, 22 June and 28 June 2021, six and four landslide disasters occurred in Youxi County and Shaowu City, Fujian Province, respectively, all of which were small-scale landslide disasters caused by local rainstorms.

Based on the actual precipitation data in Fujian Province (5 km × 5 km QPE data from China Meteorological Administration), the new model based on Random Forest and the original explicit statistical model was used to simulate the early-warning. Then, the objective forecast results of the model on 22 June and 28 were compared with the actual occurrence of landslide disasters, as shown in Table 6 and Figures 10 and 11.

According to the comparison on 22 June 2021 (Table 6, Figure 10), the six actual landslide disasters all fell in the early-warning area of the Random Forest model (one in the yellow early-warning area and six in the orange early-warning area), with a hit rate of 100%; one landslide disaster fell in the early-warning area (yellow early-warning area) of the explicit statistical model, with a hit rate of 16.7%. The landslide densities in the early-warning areas of the two models were 6.2 and 3.8 per 1000 square kilometers, respectively. The actual landslide density in the early-warning area of the Random Forest model was 1.6 times that of the explicit statistical model.

**Table 6.** Comparison between the Random Forest model and explicit statistical early-warning model.

| Actual Landslide (Number) | | 22 June 2021 Results of Different Models | | 28 June 2021 Results of Different Models | |
|---|---|---|---|---|---|
| | | Explicit Statistical Models | Random Forest Model | Explicit Statistical Models | Random Forest Model |
| | | 6 | | 4 | |
| All warning area | Accuracy (%) | 16.7 | 100.0 | 100.0 | 100.0 |
| | Number of landslides | 1 | 6 | 4 | 4 |
| | Area of warning area (km$^2$) | 262.2 | 971.9 | 4119.1 | 2359.8 |
| | Landslide density (amount of per 1000 km$^2$) | 3.8 | 6.2 | 1.0 | 1.7 |
| Yellow Alert area | Number of landslides | 1 | 1 | 2 | 0 |
| | Area of warning area (km$^2$) | 262.2 | 576.0 | 3051.3 | 1631.9 |
| | Landslide density (amount of per 1000 km$^2$) | 3.8 | 1.7 | 0.7 | 0.0 |
| Orange Alert area | Number of landslides | 0 | 5 | 2 | 4 |
| | Area of warning area (km$^2$) | 0 | 396.0 | 1067.8 | 728.0 |
| | Landslide density (amount of per 1000 km$^2$) | / | 12.6 | 1.9 | 5.5 |

According to the comparison on 28 June 2021 (Table 6, Figure 11), four landslide disasters all fell in the early-warning area (orange warning area) of the Random Forest model, with a hit rate of 100%; four landslides fell in the warning zone of the explicit statistical model (two yellow warning areas), the hit rate is also 100%. The landslide densities in the early-warning areas of the two models were 1.7 and 1.0 per 1000 square kilometers, respectively. The actual landslide density in the early-warning area of the Random Forest model was 1.7 times that of the explicit statistical model.

By comparing the results of the two models, it can be seen that the hit rate of the new model based on the Random Forest was six times that of the original model (22 June) or equivalent (28 June), and the landslide density in the early-warning area of the new model was 1.6–1.7 times that of the original. Meanwhile, it can be seen from Figure 11 that no landslide occurred in WuYiShanshi area (upper right corner of the figure frame). There is no warning area in the warning results of the Random Forest model, and there is a certain range of yellow warning areas in the warning results of the explicit statistical model warning results. The preliminary verification shows that the new model based on the Random Forest has obvious advantages with a higher hit rate, smaller warning area, and more accurate warning. Since there are few new landslide disasters in the study area, the current model verification work is relatively weak, and we will continue to track the new landslide disasters in the study area and strengthen the model validation and correction.

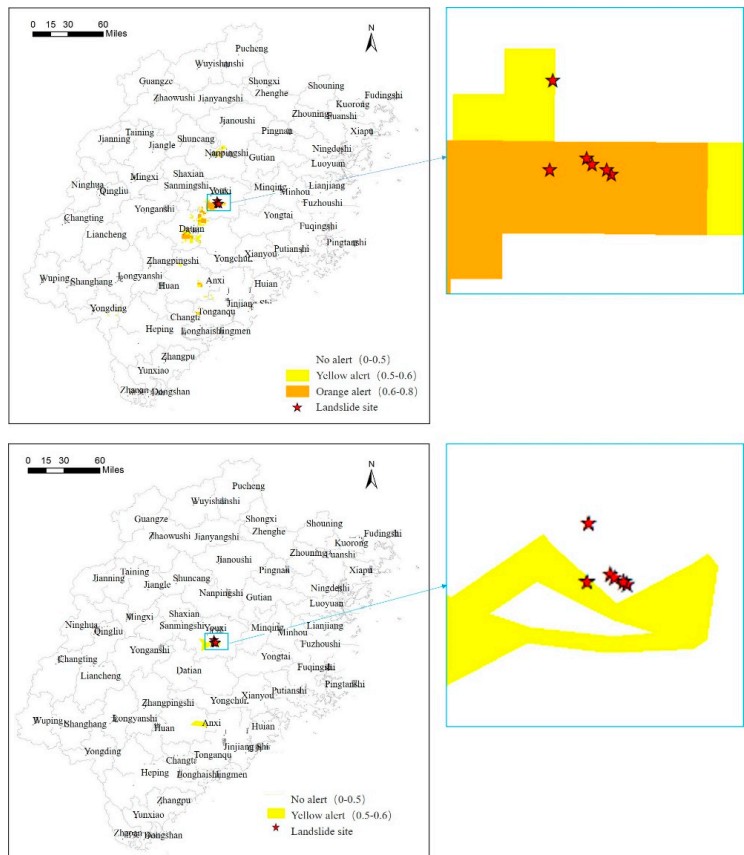

**Figure 10.** Comparison of early-warning results of different models on 22 June 2021 (above: early-warning results of the Random Forest model; below: explicit statistical model warning results).

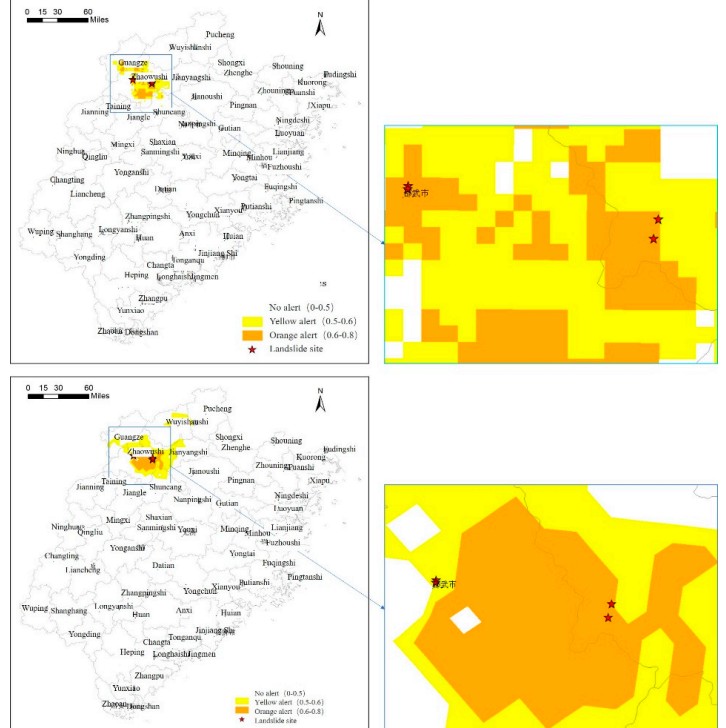

**Figure 11.** Comparison of early-warning results of different models on 28 June 2021 (above: early-warning results of Random Forest model; below: explicit statistical model warning results).

## 6. Conclusions

This paper proposed a method and process for building a regional landslide disasters early-warning model based on machine learning. Taking Fujian Province as the geological background, the landslide disaster early-warning based on machine learning in this region was carried out. Six machine learning algorithms were selected, including the Random Forest algorithm, Nearest Neighbor algorithm, Support Vector Machine, Logistic Regression, Decision Tree, Artificial Neural Network, etc. According to the topography, geological conditions, environmental conditions, human engineering activities, and other influencing factors of Fujian Province, evaluation indexes such as grade, geomorphic type, lithology, historical disasters, vegetation type, water system influence, annual rainfall, distance from house, distance from road, density of population, daily rainfall, and daily rainfall from the previous 1 to 15 days were selected. The sample data set of the early-warning model was constructed by including 27 disaster-influencing factors, and the establishment of the geological disaster early-warning model was realized through several key steps such as the construction of the sample-set, the model training and parameter adjustment optimization, and the classification of the model early-warning level. The results of the early-warning model showed that the Random Forest algorithm performed better. This algorithm has the advantages of fully applying limited samples, diversity, and accuracy; and it has the disadvantage of overfitting on certain noisy problems. In this early-warning model, the Random Forest algorithm accuracy rate was the highest (92.3%), and the model generalization ability was the best (AUC is 0.955); the second best was the Artificial Neural Network model, which has the advantages of high classification accuracy and strong learning ability, but because it is the disadvantage of requiring a large number of parameters in this model, the accuracy rate was slightly lower than that of the Random Forest model, being 0.937, and the AUC was 0.935. Contrary, the Nearest Neighbor model, Logistic Regression model, and support vector machine model have problems such as difficulty in solving multi-classification problems and requiring a large amount of training data, their AUC values being 0.924, 0.922, and 0.920, respectively. The worst model was the Decision Tree model, which has problems such as poor handling of continuous variables and easy overfitting, its AUC value being 0.904 and the accuracy 0.937. For the selection of the typical rainfall-type landslide disaster process in Fujian Province from 2019 to 2021 and for the verification of the Random Forest algorithm model, the results showed that the early-warning hit rate of the model was 100%. Compared with the early-warning results of the original explicit statistical model, the hit rate of the new model is higher than that of the original one, since, the landslide density in the early-warning area of the new model was 1.6–1.7 times higher than that of the original model. The preliminary verification showed that the new model based on the Random Forest had obvious advantages with a higher hit rate and smaller warning area, which can achieve more accurate warnings.

Research on the regional geological disasters early-warning model based on machine learning is relatively complex. Through the research in this paper, the problems of insufficient samples, limited methods, and insufficient precision in the traditional regional landslide early-warning model were solved to a certain extent. For the geological disasters early-warning model constructed by machine learning, the larger the sample data set, the higher the accuracy of the trained model, and the follow-up will increase the amount of data to optimize and improve the early-warning models.

**Author Contributions:** Data resources, J.H., Y.L., P.Z.; research methods, Y.L., P.Z.; writing—original draft preparation, Y.L., S.M.; editing, Y.L., S.M., R.X. All authors have read and agreed to the published version of the manuscript.

**Funding:** This research was financially supported by the National Natural Science Foundation of China (42077440; 41202217) and the National Key Research and Development Program of China (2018YFC15 05503).

**Institutional Review Board Statement:** Not applicable.

**Informed Consent Statement:** Not applicable.

**Data Availability Statement:** Not applicable.

**Conflicts of Interest:** There is no conflict of interest in the research methods, routes and data in this study.

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
