# Peer review of "Research on a Regional Landslide Early-Warning Model Based on Machine Learning—A Case Study of Fujian Province, China"

_forests, doi:10.3390/f13122182_

Round 1
Reviewer 1 Report
China is one of the countries with the most serious geological disasters in the world. Hence, the early warning is important in disaster prevention and mitigation. In the manuscript by Liu et al., they focus on the construction method of a regional landslide disaster early warning model based on a machine learning algorithm from the key links of training sample set construction, model training, and optimization evaluation, and model early warning. It is important and hot topic in geo-hazard study and the results could provide helpful reference for early warning model studies.
1. For the factors used in the study, why your use rainfall twice in the geological environment factors and again in rainfall-induced factors. It is unnormal in choosing factors for landslide assessment study. Please clarify it in the manuscript.
2. The words in some of the figures (i.g. Fig.2 Fig.3…) are too small to see. North Arrow and scale are missing in some of the maps. Please normalized the maps.
3. Suggest to split Fig.7 into two maps showing positive and negative samples respectively, since the points are too crowed to distinguish.
4. In the abstract, you mention, in lines 25-26, the accuracy of Random Forest algorithm performed best is 92.3%, but you mention again its accuracy as 94.3%? Is it right? Please check it and pay attention on some symbols used in the related mentioned sentences.
Reviewer 2 Report
Comments and edits are as follows:
1. The manuscript applies six kinds of machine learning algorithms to investigate the relationship between landslides and the associated parameters,taking Fujian Province, China, as the study case. It is recommended to highlight specific points of innovation. The key scientific questions addressed in this paper need further refinement.
2. “The traditional regional landslide disaster early warning model is limited by the complex landslide induction mechanism, limited data accumulation, and insufficient big data analysis methods, and has problems such as limited early warning accuracy and insufficient refinement.” In lines 11-13, the authors summarize the limitations of the subject of landslide early warning, but the core of this paper does not address this issue. Therefore, it is recommended that the scientific problem addressed should be accounted for in detail in the text.
3. Please note the difference between the content of this manuscript and the landslide susceptibility assessment, because the research content of this paper does not seem to meet the needs of landslide early warning.
4. The evaluation results obtained from the six methods need to be presented in the form of graphs, and currently only the graphs obtained from the random forest algorithm are presented.
5. Fig.2 and Fig.11 are not very clear and should be improved.
Round 2
Reviewer 2 Report
The authors conducted a case study, apply six kinds of machine learning algorithms to investigate the relationship between landslides and the associated parameters considering the geological environment, disaster-bearing body and previous rainfall. Its topic is suitable for this journal. However, according to the precision of the adopted data and the graphs showing the results (as shown partly in figure 11), it is questionable whether the accuracy of landslide warning can be achieved in the evaluation results of this manuscript.
The machine learning and statistical methods have not been strictly defined yet. In order to get the required accuracy of landslide warning, it is very fundamental to improve the data accuracy itself and consider of the landslide formation mechanism.
Finally, the innovation points are suggested to better refine in this paper.
